# Advances in Lupus Nephritis Pathogenesis: From Bench to Bedside

**DOI:** 10.3390/ijms22073766

**Published:** 2021-04-05

**Authors:** Bogdan Obrișcă, Bogdan Sorohan, Liliana Tuță, Gener Ismail

**Affiliations:** 1Department of Nephrology, Fundeni Clinical Institute, Fundeni Street 258, 022328 Bucharest, Romania; obriscabogdan@yahoo.com (B.O.); bogdan.sorohan@yahoo.com (B.S.); 2Department of Nephrology, “Carol Davila” University of Medicine and Pharmacy, 020021 Bucharest, Romania; 3Department of Nephrology, “Ovidius” University, 900527 Constanta, Romania; tutaliliana@yahoo.com

**Keywords:** lupus nephritis, systemic lupus erythematosus, pathogenesis, neutrophils, interferon, B-cells, anifrolumab, belimumab, voclosporin, Obinutuzumab

## Abstract

Systemic lupus erythematosus (SLE) is the prototype of autoimmune disorders caused by a loss of tolerance to endogenous nuclear antigens triggering an aberrant autoimmune response targeting various tissues. Lupus nephritis (LN), a major cause of morbidity and mortality in patients with SLE, affects up to 60% of patients. The recent insights into the genetic and molecular basis of SLE and LN paved the way for newer therapies to be developed for these patients. Apart from the traditional B-cell-centered view of this disease pathogenesis, acknowledging that multiple extrarenal and intrarenal pathways contribute to kidney-specific autoimmunity and injury may help refine the individual therapeutic and prognostic characterization of such patients. Accordingly, the formerly induction-maintenance treatment strategy was recently challenged with the exciting results obtained from the trials that evaluated add-on therapy with voclosporin, belimumab, or Obinutuzumab. The scope of this review is to provide an insight into the current knowledge of LN pathogenesis and future therapeutic strategies.

## 1. Introduction

Systemic lupus erythematosus (SLE) is the prototype of autoimmune disorders caused by a loss of tolerance to endogenous nuclear antigens triggering an aberrant autoimmune response targeting various tissues [1,2]. All renal compartments can be injured in SLE, but lupus nephritis (LN) is the most frequent pattern of injury encountered, affecting up to 60% of patients [3,4]. LN is associated with significant morbidity and mortality in SLE, these patients having a worse survival compared to those without nephritis, among the contributors to death being both disease-related and treatment-related factors [4,5,6]. In our experience, the 10-year and 20-year renal and/or patient survival is approximately 70% and 60%, respectively [4]. Additionally, we showed that the treatment-related morbidity is substantial, the incidence rate of infections and serious infections being 26.6 and 9.56 events per 100 patient-years, respectively [6]. Nevertheless, both patient and renal survival significantly improved over the past decades with the advent of newer treatment strategies [7,8]. Despite these advances, a substantial residual renal risk remains in LN, with up to 30% of patients progressing to end-stage renal disease [9]. After several negative trials that had failed to demonstrate the superiority of the tested drugs over the current standard of care, recent years brought ground-breaking results to the LN treatment landscape [10]. This progress was superimposed on a better understanding of the genetic and molecular basis of LN, thus paving the way for refining and individualizing patient management [1,2].

The scope of this review was to provide an insight into the current knowledge of LN pathogenesis and future therapeutic strategies.

## 2. Genetic Environment in Lupus Nephritis

SLE and LN pathogenesis clearly involve a genetic predisposition as first-degree relatives of patients with SLE are at higher risk of SLE and other autoimmune disorders [11]. In a population-based family study, Kuo et al. identified a 316-fold higher risk for twins of the patients with SLE, a 23-fold higher risk for siblings, and an 11-fold higher risk for parents, respectively, to develop autoimmune disorders (including SLE) [11]. Over the past decades, with the advent of extensive genome-wide association studies, more than 100 susceptibility loci that were linked to SLE and LN were identified [12,13]. Genes that are potentially involved in SLE pathogenesis can be broadly classified into four categories: genes involved in lymphocyte function, genes involved in innate immune signaling, genes involved in DNA clearance and complement pathway, and genes contributing to renal injury [1,12]. 

Variations in the HLA loci involved in antigen processing and presentation are among the best-characterized genetic risk factors. In LN, HLA-DR3, and HLA-DR15 alleles were associated with an increased risk for disease development, while HLA-DR4 and HLA-DR11 seemed to portend a protective effect [14]. Additionally, other HLA alleles were described as risk factors in certain subpopulations: HLA-DRB1*15 and HLA-DQB1*6:2 in Asians, HLA-DR2 in Caucasians [2]. 

In SLE and LN, the crosstalk between autoreactive B-cells and T-cells is essential. As B-cells have a dual function being involved in both antigen presentation, in association with dendritic cells, and autoantibody production, several SLE-associated genes affecting B-cell receptor (BCR) function and intracellular signaling (such as BANK1, RasGRP3, the Src-family associated tyrosine kinases-LYN/BLK/CSK-, TLR7-9, IRF5, IRF7) have been linked to the disease pathogenesis [1]. Immunogenic DNA and RNA could, either alone or in the form of immune complexes, activate the B-cells, via BCR and FcγRII, and immature dendritic cells, via FcγRII that will further activate several intracellular pathways (involving NF-κB, interferon regulatory factors IRF5 and IRF7) resulting in increased cell survival, pro-inflammatory cytokines and type I interferon production, and, ultimately, autoantibody synthesis [1,2,12]. Additionally, the nuclear autoantigens can, through endocytosis of immune complexes, activate the endosomal Toll-like receptor (TLR7-9) that will recruit several transcriptional pathways, such as NF-κB, IRF5, and IRF7, that will augment the cytokine production [1,2,12]. In addition to DNA and RNA endosomal sensors, an important role of cytosolic sensors in the pathogenesis of LN, such as RIG1/MDA-5 and cGAS-STING, must also be highlighted (Figure 1) [1]. Signaling through type I IFN (interferon) receptor (IFNαRII) is a molecular hallmark of SLE and LN, and several single-nucleotide polymorphisms related to the genes involved in the downstream signal transduction pathways (STAT4, IRF5, IRF7) seem to be involved in the expression of IFN stimulated genes (ISGs) affecting key functions of innate and acquired immunity and, possibly, of resident renal cells [12,15]. Furthermore, T-cells have several important roles in SLE pathogenesis, especially T-follicular-helper-cells that stimulate B-cell autoreactivity and autoantibody production and IL-17-producing T-helper-cells (Th17-cells) that seem to be an essential driver of LN [1]. Overall, these genetic abnormalities are important contributors to the generation of the SLE “cytokine signature” milieu comprising mediators such as type I interferons, B-cell survival factors (BAFF), and several interleukins (IL-6, IL-12, IL-17, IL-23), all potential treatment targets (Figure 1 and Table 1).

Lastly, genes involved in DNA clearance and complement system activity were essential contributors to SLE and LN pathogenesis [2,12,15]. Particularly, genes encoding DNase I, DNase III, DNase-γ involved in DNA clearance, or genes encoding C1q/C4 complement factors were shown to be significantly associated with SLE development [1]. Neutrophils were key players in SLE pathogenesis through a specific form of cellular death (NETosis) in which neutrophil extracellular traps (NETs) containing chromatin were released [29]. Patients who were deficient in either DNAase I, involved in neutrophil extracellular traps (NETs) degradation, and C1q, essential for NETs opsonization, and clearance, had a continuous source of nuclear autoantigens capable of sustaining the autoimmunity [14].

## 3. Role of Interferon Signaling in Lupus Nephritis

The type I interferon system is an essential component of innate and adaptive immunity that, for decades, was regarded as the main line of defense against viral infections [8,15]. However, the type I IFN system has additional important roles in the pathogenesis of autoimmune disorders, especially in SLE and LN. Patients with SLE have an increased expression of type I IFN stimulated genes in peripheral blood leukocytes, while patients with LN show higher IFN scores, especially during active renal disease [30]. Additionally, the molecular profiling of the kidney biopsy tissue in patients with LN may help predict treatment response as ISGs are upregulated in a LN flare and are associated with a complete renal response, while the complement genes are upregulated in the renal tissue of non-responders, suggesting that both pathways are involved but have distinct roles in the pathogenesis of LN [31].

Many cell types can synthesize type I IFN, but plasmacytoid dendritic cells (pDCs) are the master IFN-producing cells [15]. During the normal antiviral response, intracellular activation of Toll-like receptors 7/9 will stimulate several transcription pathways that eventually will inhibit the viral replication at several stages [15]. In patients with SLE and LN, there is a characteristic IFN “signature” due to the increased expression of ISGs that, in turn, coordinates several inflammatory responses and promotes kidney-specific autoimmunity [2,14]. The driver of IFN-α in LN are the interferogenic immune complexes (ICs) containing nucleic acids (DNA/RNA). These ICs will be internalized through FcγRII on the pDCs surface, reached endosomal DNA/RNA sensors (TLR7/9) or cytosolic sensors (RIG1/MDA-5 and cGAS-STING), and activate downstream signaling pathways (NF-κB, IRF5, and IRF7) resulting in the increased production of type I IFN and other pro-inflammatory cytokines [1]. The role of type I IFN in sustaining autoimmunity in LN involves the stimulation of antigen presentation by DCs (dendritic cells) and macrophages, the enhancement of the survival of autoreactive T- and B-cells, the activation of B-cell differentiation and class-switching generating memory plasma cells and autoantibody production, the suppression of regulatory T-cells [1,15]. Nucleic acids can also induce type I IFN production in intrarenal DCs and some intrinsic renal cells (mesangial cells, endothelial cells), the ultrastructural hallmark of enhanced intrarenal IFN signaling being the presence of tubuloreticular inclusions in glomerular endothelial cells [32,33].

Recently, a role of type III IFN (IFN-λ) for stimulating immune dysregulation and tissue inflammation has been proposed in a murine model of lupus [34]. In this TLR7-induced lupus murine model an increased production of IFN-λ was observed, while the deficiency of its receptor (IFN-λR1) significantly decreased the activation of not only the immune cells, but also of the keratinocytes and mesangial cells [34]. Additionally, mesangial cells seemed to directly respond to IFN-λ with the upregulation of ISGs and the development of lupus-associated renal pathology [34]. Nonetheless, IFN-α and IFN-λ may have distinct, yet complementary, effects on various cells involved in the pathogenesis of SLE and LN [34]. 

Recently, a role for plasma membrane TLR in the pathogenesis of LN was suggested in a patient with class V LN carrying a rare TLR1 variant [35]. As TLR1 is expressed on the podocyte’s surface, TLR ligands might contribute to direct podocyte injury and initiation of renal pathology [35]. Thus, both endosomal and plasma membrane TLR seem to be involved in the LN pathogenesis, and the activation of different TLR might explain the clinical heterogeneity of LN. 

The pathogenic relevance of type I IFN system was explored clinically as anifrolumab, a type I IFN receptor antagonist, substantially reduced disease activity in patients with moderate to severe SLE, with a greater effect in those with high IFN signature at baseline [36]. Currently, an ongoing trial (TULIP-LN) is evaluating the efficacy of anifrolumab in patients with LN [14].

## 4. Potential Specific Antigens in Lupus Nephritis

Renal injury in LN occurs through either circulating immune complex deposition, autoantibodies binding to “planted” glomerular antigens, or in situ immune complex formation within glomeruli [14]. Despite that several classical autoantigens have been associated with proliferative LN (double-stranded DNA, nucleosomes, U1 small nuclear ribonucleoprotein, etc.), there is an increasing evidence over the past decade that distinct nephropathic autoantigens might be associated with different classes of LN [33]. Accordingly, autoantibodies targeting glomerular Annexin A2, a multifunctional protein that belongs to a family of Ca2+-regulated phospholipid-biding proteins, and Moesin, which is part of a protein complex involved in Rho GTPase signaling that is essential for actin cytoskeleton remodeling, cell adhesion, and motility were shown to be associated with proliferative, but not membranous, LN [37,38]. Distinctly, autoantibodies targeting NCAM1 (neural cell adhesion molecule 1), a member of the immunoglobulin superfamily of proteins that is expressed on podocytes, were identified in 6.6% of cases of membranous LN [39]. Recently, glomerular subepithelial deposits containing exostosin 1 and exostosin 2 (EXT1/2) were identified in patients with PLA2R-negative and THSD7A-negative membranous nephropathy [40,41]. Over 80% of patients showing positive staining for EXT1/2 had clinical or histological features suggestive of autoimmunity, mostly SLE, such as antinuclear antibodies, anti-dsDNA antibodies, tubuloreticular inclusions. Additionally, among patients with membranous LN, the presence of exostosin-positivity was associated with a distinct clinical phenotype, these patients being younger, having fewer chronicity features on kidney biopsy, and an overall better prognosis than exostosin-negative patients [42]. Although not fully validated, the concept that different antigens might drive different clinical and histological phenotypes of LN is appealing.

## 5. Cellular Players in Lupus Nephritis—The Role of Neutrophils and Tertiary Lymphoid Organs

For many years, the understanding of SLE and LN pathogenesis was centered around cells of adaptive immunity (DCs, T- and B-cells). Nonetheless, neutrophils emerged as important players in SLE and LN pathogenesis, the expression of neutrophil-specific genes being prevalent and correlating with disease activity [43]. Apart from the molecular point of view, their clinical relevance in LN is illustrated by the separate quantification of neutrophils and karyorrhexis (representing apoptotic cell death of neutrophils) in the recently modified NIH (National Institutes of Health) activity index [44]. Moreover, the contribution of these cells to the extent of intraglomerular inflammation is relevant to the long-term renal outcome. We identified, in a cohort of patients with LN, that, irrespective of LN class, the extent of crescent formation was an independent predictor of poor prognosis (HR, 1.068 for each percentage of glomeruli with crescents; 95% CI 1.003 to 1.136) [4]. The concept that the type of histological lesion is a more important predictor for the renal outcome than the histological class is gaining popularity in LN, this shift in paradigm being driven by the better understanding of the cellular and molecular basis of LN pathogenesis [4]. 

Neutrophils are the first line of defense mechanism against infections, but also important contributors to autoimmunity through a unique form of cell death called NETosis (neutrophil extracellular trap) [43,45]. NETs, which are extracellular DNA structures covered by neutrophil antimicrobial peptides (LL-37 and HNP-human neutrophil peptide), are abundantly released by neutrophils of SLE patients and represent the link between these cells and the IFN-α signaling in SLE and LN [43,44,46]. Hakkin et al. showed that a subset of patients with SLE had an impairment of NETs degradation by the endonuclease DNase 1, thus becoming potentially pathogenic [47]. Indeed, at least two potential mechanisms for impaired NET degradation were proposed, the presence of DNase 1 inhibitors or anti-NET antibodies that can confer protection from DNase-mediated degradation, and this phenomenon correlated with renal injury [47]. Additionally, the NETs are the underlying molecular basis for the positive feedback loop engaged by the interaction of neutrophils and pDCs [43]. Garcia-Ramo et al. showed that SLE NETs contain, in addition to self-DNA structures, large amounts of the antimicrobial peptide LL37, which is essential for the immunogenicity of self-nucleic acids, thus facilitating their uptake by pDCs and production of IFN-α in a DNA- and TLR-9-dependent manner [48]. Moreover, the release of NETs by neutrophils was induced after IFN-α priming and anti-ribonucleoprotein antibody exposures creating the premises for the perpetual amplification between the IFN-primed neutrophils and NET-stimulated IFN production by pDCs [48]. Additionally, Lande et al. showed that SLE patients developed autoantibodies against both self-DNA and neutrophil antimicrobial peptides (LL-37 and HNP) [49]. These anti-LL-37 and anti-HNP autoantibodies seem to have a dual role in SLE, first, by facilitating the FcγRII-mediated endocytosis of self-DNA-antimicrobial peptides by pDCs and further stimulating the TLR9-mediated type I IFN release, and second, by enhancing the NET release from IFN-primed neutrophils [43,49]. Moreover, the immunogenic self-DNA-antimicrobial peptides engage the B-cell receptor and TLR9 activating the autoreactive B-cells to synthesize anti-LL-37 and anti-HNP autoantibodies further promoting the chronic pDCs activation and sustaining autoimmunity in SLE [43,49]. This interaction between neutrophils, pDCs, and autoreactive B-cells provides the rationale for the suppression of NET formation, interference with TLR or IFN signaling, as potential therapeutic targets.

In SLE, B-cells are important contributors to both systemic and local immunity and inflammation. The diversification of B-cell clones occurs after autoantigen exposure and in SLE, the B-cell receptor (BCR) “repertoire”, as an expression of autoantibody response to multiple autoantigens, is distinct from other immune-mediated disorders [50]. Bashford-Rogers et al. observed that, in SLE, the B-cells showed an increased dysregulation of BCR repertoire and an abnormal isotype-specific clonal diversity consistent with the involvement of multiple autoantigens [50]. Unexpectedly, the IgA isotype was largely responsible for this increase in clonality suggesting that microbial antigens might be important drivers of autoimmunity in SLE and consistent with previous observations that the IFN signaling seen in SLE closely resembles the immune response to viral infections [50]. Additionally, a distinct response of B-cells subpopulations to different immunosuppressive agents was seen: the isotype-switched and clonally expanded B-cells persisted after rituximab treatment but were reduced after mycophenolate mofetil [50]. Apart from the well-defined systemic effect, organized intrarenal B-cell infiltrates are increasingly recognized as important contributors to local autoimmunity and inflammation [51]. These B-cell infiltrates are encountered in over 50% of patients with LN and are structured into four increasingly organized levels of aggregates in the tubulointerstitium [52,53,54]. The presence of these tertiary lymphoid structures correlates with the severity of both glomerular and tubulointerstitial lesions, their pathogenic role being to enhance the local immune response, including local antibody production, antigen presentation, and proinflammatory cytokine production [52,53,54]. Moreover, as tubulointerstitial lesions are not adequately captured in the current ISN/RPS (International Society of Nephrology/Renal Pathology Society) classification, their significance to long-term renal prognosis should be highlighted [4]. As an example, we identified that the presence of tubulitis was associated with a 13.1-fold higher risk of a worse outcome (HR, 13.1; 95% CI 1.3 to 131) [4].

The contribution of intrarenal B-cells to kidney-specific autoimmunity is clinically relevant as the failure of rituximab to increase the renal response rates was due to an incomplete depletion of peripheral and, possibly, interstitial B-cells [55]. In this regard, obinutuzumab determines a superior peripheral B-cell depletion compared to rituximab, and its efficacy on tissue depletion is currently being evaluated by a repeat kidney biopsy at 52 weeks in the NOBILITY trial [8,10] (Table 1).

## 6. From Antibody to Injury in Lupus Nephritis—The Role of the Complement System

The complement system has a dual role in the pathogenesis of SLE, especially LN. Firstly, the complement has a protective role against the development of autoimmunity through opsonization and enhancement of nuclear material and apoptotic debris removal [28,56]. This is illustrated in patients with genetic deficiency of C1q or C4 who developed lupus or a lupus-like disease [56]. A distinct situation is the presence of anti-C1q antibodies, more prevalent in LN compared to nonrenal SLE, that seem to confer an acquired amplification loop of the classical pathway of complement activation [56,57]. Secondly, the complement activation through both classical and alternate pathways is an important mediator to autoantibody-mediated renal injury [56]. A murine model of complement factor H deficiency outlines the importance of the alternate pathway for LN development [58]. The complement-targeted therapies significantly evolved in the past years, with several trials in SLE and LN being currently ongoing (Table 1).

## 7. Transition from Bench to Bedside—Novel Therapies in Lupus Nephritis

Although there has been a significant improvement in both patient and renal survival in patients with LN with the advent of cyclophosphamide-based and mycophenolate mofetil-based treatment regimens, there is still a substantial treatment-related morbidity and mortality [6]. We have shown, in a study that included 101 patients with severe proliferative LN, that an initial high-dose oral corticosteroid regimen (≥0.5 mg/kg/day in the first month of induction therapy) increased the risk for serious infections by 7.5-fold (HR, 7.57; 95% CI 1.64–34.8) [6]. Accordingly, continuous efforts have been made to refine the immunosuppressive regimens, e.g., by limiting cyclophosphamide and corticosteroids exposure, by switching to CD20-depleting or multitarget-based regimens [6]. After decades of negative trials, in which several new agents added to the current standard of care regimens (both cyclophosphamide-based and mycophenolate mofetil-based) failed to show an improvement in the treatment response rates, recent years brought ground-breaking results to the LN treatment [10]. The progress that has been made in the understanding of LN pathogenesis created the premises for new therapeutic agents to be developed. Currently, several molecular pathways are being explored as potential therapeutic targets in LN, thus offering the possibility to better individualize patient management (Figure 2 and Table 1).

Molecular targets that are currently or were evaluated in clinical trials can be broadly classified into: inflammatory mediators (IFN, IL-6, IL-12, IL-17A, IL-23), co-stimulation blockade (CD80/86:CD28 and CD40L:CD4), B-cell survival factors inhibition, direct B-cell or plasma cell depletion, and complement inhibition (Figure 2 and Table 1). Despite that some of these newer agents did not consistently improve clinical outcomes, several recent trials showed positive results that may shift the treatment paradigm of LN and raise the possibility for these agents to enter the clinical practice as first-line options [1,10].

Voclosporin, a new-generation calcineurin inhibitor (CNI), has a similar potency to inhibit the activity of T-cells as cyclosporine or tacrolimus, but without the need to monitor the trough levels and with a better metabolic profile. In a phase II trial (AURA trial), voclosporin (23.7 mg or 39.5 mg, twice daily) was evaluated as part of a multitarget regimen comprising mycophenolate mofetil (MMF, 2 g/day) and a rapidly tapered low-dose oral corticosteroids (initial dose of 20–25 mg/day, tapering to 5 mg/day by week 8 and to 2.5 mg/day by week 16) [22]. In this trial, the low-dose voclosporin regimen had significantly higher complete renal response rates compared to the placebo arm at both 24 (32.6% vs. 19.3%) and 48 weeks (49.4% vs. 23.9%), respectively [22]. Recently, the positive results of the phase III AURORA trial were communicated at scientific meetings which confirmed that voclosporin (23.7 mg, twice daily) as an add-on therapy to MMF and low-dose steroids increased the efficacy of the induction regimen with an effect size of 18.5% for a complete renal response [10,22].

Though the initial enthusiasm for depleting CD20+ cells was reduced after the negative results of the LUNAR trial [25], the blockade of humoral immunity regained its popularity after belimumab and obinutuzumab trials reported positive results in proliferative LN [10]. Belimumab, a recombinant human IgG1λ monoclonal antibody that inhibits B-cell activating factor, showed efficacy and safety as an add-on therapy to steroids plus either MMF or cyclophosphamide-azathioprine regimens when given intravenous monthly over a period of 104 months, with an effect size of 11% for a PIRR (primary efficacy renal response) [10,23]. In the phase II CALIBRATE trial, the addition of belimumab to a rituximab/cyclophosphamide/steroid regimen impaired the maturation of naïve B-cells and stimulated the negative selection of autoreactive B-cells [24]. However, this was more of a proof-of-concept trial to evaluate the effect of belimumab after B-cell depletion with rituximab and neither designed nor powered to evaluate the superiority in terms of enhancing clinical response rates [24]. Accordingly, the reduction in the percentage of naïve B-cells after belimumab addition supports the dependence on BAFF of the differentiation of transitional B-cells to naïve B-cells [24].

Obinutuzumab, a humanized type II anti-CD20 monoclonal antibody, determines a superior B-cell depletion compared to rituximab [59]. In the phase II NOBILITY trial, Obinutuzumab given intravenously on day 1 and 14, with a repeated administration after 6 months, over a background standard-of-care consisting of MMF and steroids showed superiority in terms of achieving a complete renal response with an effect size of 22% at week 76 [10]. Currently, Obinutuzumab is further evaluated in the phase III REGENCY trial in patients with proliferative LN (Table 1) [10]. Daratumumab, a human monoclonal antibody that targets CD38, which determines plasma cell depletion and is currently approved for the treatment of multiple myeloma, induced clinical remission in two cases of refractory life-threatening lupus [27]. Thus, plasma cell depletion by daratumumab or proteasome inhibitors (bortezomib, ixazomib) is appealing as a treatment option for LN [1].

Targeting the IFN signaling through anifrolumab, a human monoclonal antibody to type I IFN receptor subunit 1, was associated with a higher percentage of clinical responses in patients with moderate-to-severe SLE, especially in those with a high interferon gene signature [16,36]. Currently, the phase II TULIP-LN1 trial evaluates the efficacy of anifrolumab in patients with proliferative LN (Table 1).

In summary, these recent insights into the genetic and molecular basis of SLE and LN paved the way for newer therapies to be developed for these patients. Apart from the “traditional” B-cell-centered view of this disease pathogenesis, acknowledging that multiple extrarenal and intrarenal pathways contribute to kidney-specific autoimmunity and injury may help refine the individual therapeutic and prognostic characterization of such patients.

## Figures and Tables

**Figure 1 ijms-22-03766-f001:**
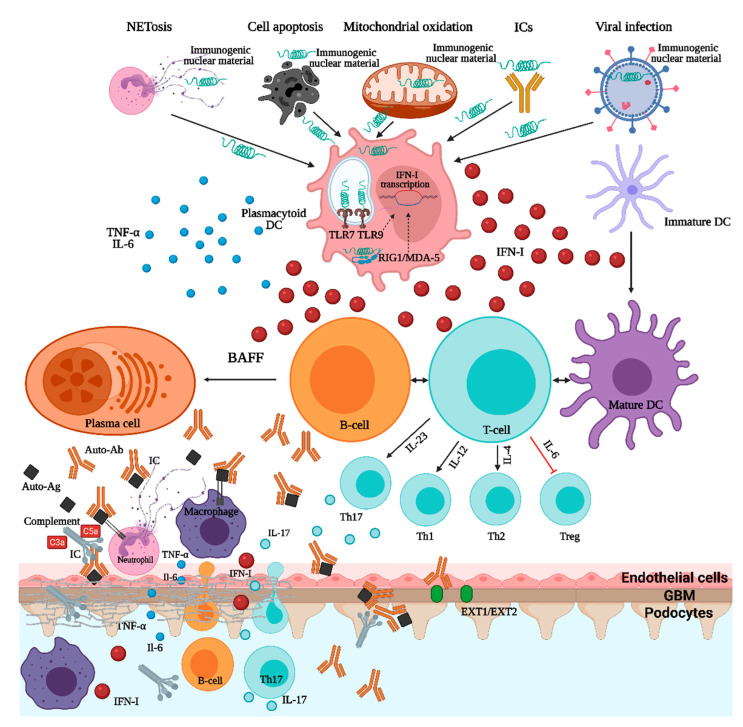
Pathogenesis of lupus nephritis. ICs—immune complexes; TNF—tumor necrosis factor; IL—interleukin; DC—dendritic cell; IFN—interferon; TLR—toll-like receptor; RIG1/MDA-5—retinoic acid inducible gene 1/melanoma differentiation-associated protein 5; BAFF—B-cell activating factor; Ab—antibody; Ag—antigen; Th—T-helper; EXT1/EXT2—exostosin1/exostosin2.

**Figure 2 ijms-22-03766-f002:**
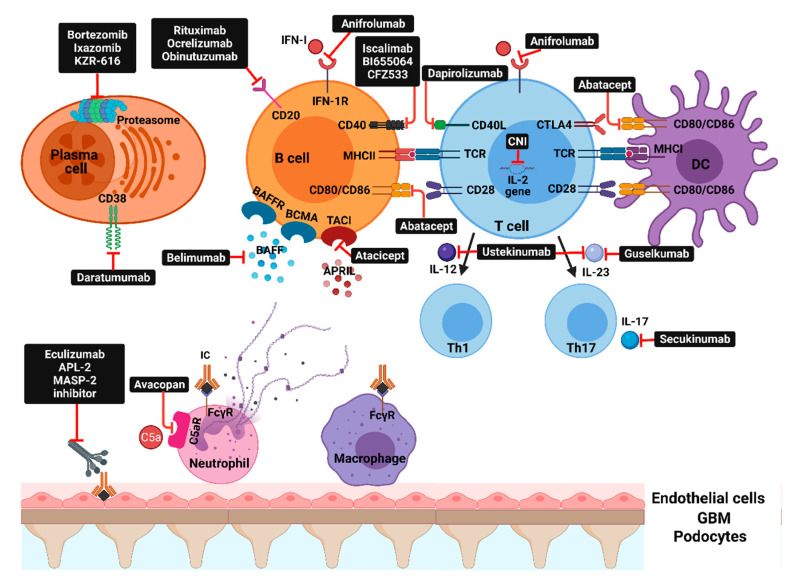
Molecular pathways and corresponding therapeutic agents: IFN-I—interferon type I; IFN1-R—type 1 interferon receptor; BAFFR—B-cell activating factor receptor; BCMA—B-cell maturation antigen; TACI—transmembrane activator and calcium-modulator and cyclophilin ligand interactor; CD40L—CD40 ligand; CTLA4—cytotoxic T-lymphocyte antigen 4; CNI—calcineurin inhibitors; TCR—T-cell receptor; MHC—major histocompatibility complex; DC—dendritic cell; BAFF—B-cell activating factor; APRIL—a proliferation-inducing ligand; IL—interleukin; Th—T-helper; APL-2—pegcetacoplan; MASP-2—mannan-binding lectin serine protease 2; C5aR—complement C5aR; IC—immune complex; FcүR—Fc-gamma receptors.

**Table 1 ijms-22-03766-t001:** Molecular pathways and corresponding agents tested in clinical trials in SLE and LN.

Therapeutic Agent	Molecular Target	Trial Name	Trial Phase	Results	Trial Reference
**Anifrolumab**	IFN	TULIP-2 TULIP-LN1	III II	Positive Ongoing	[16] NCT02547922
**Secukinumab**	IL-17A	SELUNE	III	Ongoing	NCT04181762
**Ustekinumab**	IL-12/IL-23	-	II	Positive	[17,18]
**Sirukumab**	IL-6	-	II	Endpoint not met	[19]
**Guselkumab**	IL-23	ORCHID-LN	II	Ongoing	NCT04376827
**Dapirolizumab**	CD40L	PHOENYCS GO	III	Ongoing	NCT04294667
**Iscalimab**	CD40	-	II	Ongoing	NCT03610516
**BI 655064**	CD40	-	II	Ongoing	NCT03385564 NCT02770170
**Abatacept**	CD28-CD80	ACCESS	III	Endpoint not met	[20]
**Atacicept**	BAFF/APRIL	-	II/III	Terminated early due to unanticipated safety issues	[21]
**Voclosporin**	Calcineurin	AURA AURORA	II III	Positive Positive (results to be published)	[22] NCT03021499
**Belimumab**	BAFF	BLISS-LN CALIBRATE	III II	Positive Endpoint not met	[23] [24]
**Blisibimod**	BAFF	CHABLIS7.5 CHABLIS-SC2	III III	Withdrawn Withdrawn	NCT02514967 NCT02074020
**Rituximab**	CD20	LUNAR	III	Endpoint not met	[25]
**Ocrelizumab**	CD20	-	III	Stopped early due to higher number of serious infections with ocrelizumab	[26]
**Obinutuzumab**	CD20	NOBILITY REGENCY	II III	Positive (results to be published) Ongoing	[10] NCT04221477
**Daratumumab**	CD38	-	-	Therapy for consideration	[27]
**Ianalumab**	BAFF receptor B-cells	-	-	Therapy for consideration	-
**Bortezomib**	Plasma cell	-	IV	Withdrawn	NCT01169857
**Ixazomib**	Plasma cell	-	I	Insufficient enrolment	NCT02176486
**Eculizumab**	C5	-	-	Therapy for consideration	[28]
**Ravulizumab**	C5	-	II	Ongoing	NCT04564339
**Avacopan**	C5a	-	-	Therapy for consideration	-
**Narsoplimab**	MASP-2	-	II	Ongoing	NCT02682407
**APL-2**	C5	-	II	Ongoing	NCT03453619
**LNP023**	Factor B	-	II	Therapy for consideration	-

## Data Availability

The data presented in this study are available in this review “Advances in Lupus Nephritis Pathogenesis: From Bench to Bedside”.

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
