# Peer review of "Advances in Lupus Nephritis Pathogenesis: From Bench to Bedside"

_ijms, 2021, doi:10.3390/ijms22073766_

Round 1
Reviewer 1 Report
good detailed overview. really no changes except minor spellcheck etc.
Author Response
We are submitting the reply to the composition comments you have made on our manuscript entitled “Advances in lupus nephritis pathogenesis: From bench to bed-side?” coauthored by Bogdan ObriÈ™că, Bogdan Sorohan, Liliana Tuță and Gener Ismail.
We have revised the manuscript based on the comments made by the reviewers.
Together with revised manuscript here is our answer to the reviewer’s comments.
Reviewer 1
good detailed overview. really no changes except minor spellcheck etc.
Thank you. We have revised the entire manuscript for spelling and grammatical errors and corrected as appropriately.
We hope that we have addressed all the issues of your comments.
Sincerely Yours,
Gener Ismail MD, PhD
Corresponding author: Gener Ismail MD – Department of Nephrology, Fundeni Clinical Institute,
258 Fundeni Street, District 2, Bucharest, Romania, zip code 022328; gener.ismail@umfcd.ro
Reviewer 2 Report
Dr. Obrișcă and co-authors describes the latest insights into our current knowledge on the lupus nephritis pathogenesis and its therapies. This work is a good-organized and very-well written review. It was a great pleasure to read this manuscript. Overall, I have no complains about this work, just some minor suggestions:
- Figure 1: please make text readable at the top of the Figure, where it says "Immunogenic nuclear material", etc.
- Figure 2: please make the font bigger across the figure.
- Page 2, lines 48-50: it is not very clear what you want to say here. It is difficult to understand what numbers "316", "23" and "11" mean. Please correct.
- Adding a paragraph on the role of stimulator of interferon genes (STING) in LN would strength the work even more.
Author Response
We are submitting the reply to the composition comments you have made on our manuscript entitled “Advances in lupus nephritis pathogenesis: From bench to bed-side?” coauthored by Bogdan ObriÈ™că, Bogdan Sorohan, Liliana Tuță and Gener Ismail.
We have revised the manuscript based on the comments made by the reviewers.
Together with revised manuscript here is our answer to the reviewer’s comments.
Reviewer 2
Dr. Obrișcă and co-authors describes the latest insights into our current knowledge on the lupus nephritis pathogenesis and its therapies. This work is a good-organized and very-well written review. It was a great pleasure to read this manuscript. Overall, I have no complains about this work, just some minor suggestions:
- Figure 1: please make text readable at the top of the Figure, where it says "Immunogenic nuclear material", etc.
Thank you for the observation. We have revised the entire figure so it is more readable.
- Figure 2: please make the font bigger across the figure.
Thank you for the observation. We have also revised this figure.
- Page 2, lines 48-50: it is not very clear what you want to say here. It is difficult to understand what numbers "316", "23" and "11" mean. Please correct.
Thank you for this comment. Indeed it is a bit confusing so we have rewrote the sentence accordingly to be more clear of our meaning.
- Adding a paragraph on the role of stimulator of interferon genes (STING) in LN would strength the work even more.
Thank you for the suggestion. We have added a reference to STING.
We hope that we have addressed all the issues of your comments.
Sincerely Yours,
Gener Ismail MD, PhD
Corresponding author: Gener Ismail MD – Department of Nephrology, Fundeni Clinical Institute,
258 Fundeni Street, District 2, Bucharest, Romania, zip code 022328; gener.ismail@umfcd.ro
Reviewer 3 Report
This review looks nice, however does not add anything new to the available literature. please see these two published reviews for instance;
1- https://www.ncbi.nlm.nih.gov/pmc/articles/PMC7405261/
2- https://www.annualreviews.org/doi/10.1146/annurev.med.52.1.63
Author Response
We are submitting the reply to the composition comments you have made on our manuscript entitled “Advances in lupus nephritis pathogenesis: From bench to bed-side?” coauthored by Bogdan ObriÈ™că, Bogdan Sorohan, Liliana Tuță and Gener Ismail.
We have revised the manuscript based on the comments made by the reviewers.
Together with revised manuscript here is our answer to the reviewer’s comments.
Reviewer 3
This review looks nice, however does not add anything new to the available literature. please see these two published reviews for instance;
Thank you for your comments. Although we are aware that some information is repeatedly highlighted in other papers as well (but this is due to their relevance in this disease pathogenesis), we strongly think that our review emphasizes provides the latest updates in the pathogenesis of SLE and LN, with a focus on lupus nephritis, not accurately captured so for (e.g.: latest studies evaluating genetic background; novel nephropathic antigens that were only recently published and their potential significance; the role of neutrophils; the important role not only of type I IFN, but also of other IFN types, which was not previously captured; the significance of other types of TLR not only the endosomal types). Additionally, we tried to put these novel findings in the context of the most recent trials.
We hope that we have addressed all the issues of your comments.
Sincerely Yours,
Gener Ismail MD, PhD
Corresponding author: Gener Ismail MD – Department of Nephrology, Fundeni Clinical Institute,
258 Fundeni Street, District 2, Bucharest, Romania, zip code 022328; gener.ismail@umfcd.ro
Round 2
Reviewer 3 Report
Thank you for your clarification. However, the author should also clarify their contribution to the MS. I highly recommend mentioning previous publications (at least the review papers) in this domain and elaborate on your contribution. The aim is not to survey the literature, but to elaborate on your contribution.
Author Response
We are submitting the reply to the composition comments you have made on our manuscript entitled “Advances in lupus nephritis pathogenesis: From bench to bed-side?” coauthored by Bogdan ObriÈ™că, Bogdan Sorohan, Liliana Tuță and Gener Ismail.
We have revised the manuscript based on the comments made by the reviewers.
Together with revised manuscript here is our answer to the reviewer’s comments.
Thank you for your clarification. However, the author should also clarify their contribution to the MS. I highly recommend mentioning previous publications (at least the review papers) in this domain and elaborate on your contribution. The aim is not to survey the literature, but to elaborate on your contribution.
Thank you for your comments. Indeed, our work in the field of lupus nephritis is captured in references 4 and 6. We have further detailed our findings throughout the manuscript as appropriately. Additionally, I would also like to highlight the reference 45, that is a case based review focusing on a series of patients with hypocomplementemic urticarial vasculitis, that focuses on the clinical relevance of anti-C1q antibodies in autoimmune disorders, their importance in LN pathogenesis being highlighted in the manuscript.
We hope that we have addressed all the issues of your comments.
Sincerely Yours,
Gener Ismail MD, PhD
Corresponding author: Gener Ismail MD – Department of Nephrology, Fundeni Clinical Institute,
258 Fundeni Street, District 2, Bucharest, Romania, zip code 022328; gener.ismail@umfcd.ro
Round 3
Reviewer 3 Report
Unfortunately, you did not address my concerns or maybe it was a misunderstanding. As I mentioned before, I do not find the difference between the previously published review (https://www.ncbi.nlm.nih.gov/pmc/articles/PMC7405261/) that recently published and yours.
I asked you please describe the difference between your manuscript and the review that I mentioned. I'm afraid it would be a duplicate publication.
Author Response
We are submitting the reply to the composition comments you have made on our manuscript entitled “Advances in lupus nephritis pathogenesis: From bench to bed-side?” coauthored by Bogdan ObriÈ™că, Bogdan Sorohan, Liliana Tuță and Gener Ismail.
We have revised the manuscript based on the comments made by the reviewers.
Together with revised manuscript here is our answer to the reviewer’s comments.
Reviewer 3
Unfortunately, you did not address my concerns or maybe it was a misunderstanding. As I mentioned before, I do not find the difference between the previously published review (https://www.ncbi.nlm.nih.gov/pmc/articles/PMC7405261/) that recently published and yours.
I asked you please describe the difference between your manuscript and the review that I mentioned. I'm afraid it would be a duplicate publication.
Our answer:
Regarding the review that you just mentioned, there are several important and recent information that are not accurately highlight in this or other reviews:
- We provide an updated picture of the genetic predisposition in lupus nephritis, not captured at all in the review mentioned.
- We detailed the different roles of IFN system in LN, not only reflecting the contribution of IFNα, but also the contribution of IFNλ (its in only very recently recognized to have a very significant role in LN pathogenesis).
- We detailed the roles of different TLR, not only the endosomal TLR (7-9), but also the potential contribution of plasma membrane TLR in different subsets of patients with LN
- We provided a separate discussion on the nephropatic antigens in LN and their involvement in different LN classes. This topic is not mentioned at all in the review that you proposed. Additionally, the recognition of this distinct nephropatic antigens was only recently acknowledged. We mentioned not only the role of Annexin 5, but also of Moesin, NCAM1 and exostosin. We highlighted that different antigens might drive different LN classes.
- In addition to the glomerular pathology, we also discussed on the important role of tubulointerstitial lesions. Tertiary lymphoid organs are important drives of local autoimmunity and this part is not captured in the review that you mentioned. Additionally, we have shown that in our experience tubulointerstitial lesion are important predictors of long-term renal outcome.
- We also discussed some recent data showing the distinct BCR repertoire and how this might influence the treatment response. Again, this is not touched in the review that you mention.
- Lastly, all the information that is structured in this review is a reflection of our vast clinical experience with LN patients and of how we translate the pathogenesis into the current management of such patients. We have detailed accordingly our studies and put them in the context of the review.
- We strongly believe that our review is quite distinct from the one that you suggested to us and could bring important additional information regarding pathogenesis and treatment of LN.
We hope that we have addressed all the issues of your comments.
Sincerely Yours,
Gener Ismail MD, PhD
Corresponding author: Gener Ismail MD – Department of Nephrology, Fundeni Clinical Institute,
258 Fundeni Street, District 2, Bucharest, Romania, zip code 022328; gener.ismail@umfcd.ro